# DynaVect: Context-Aware Modulation of Global Edit Directions for Controllable GAN Editing

## Abstract

Text-guided editing of generative models like StyleGAN has become a popular method for image manipulation. Current approaches face a trade-off. Optimization-based methods produce edits that are too subtle. This fails to meet the user's intent for changes. On the flip side, methods that use a single global edit vector often cause unwanted attribute entanglement and identity loss. In this work, we propose DynaVect, a hybrid approach that attempts to resolve this trade-off. Our approach is a lightweight Dynamic Contextual Modulator. The DCM is a network trained to predict a personalized correction (or delta), based on the source image's features. At inference time, this learned delta is used to steer the global edit direction. This results in changes that are visually different while attempting to preserve the original identity. We train our modulator using an optimization-distillation technique. This technique involves creating a fast feed-forward model that approximates the quality of slow, per-image optimization. We demonstrate that our method produces qualitatively superior results that better align with users expectations as compared to traditional metrics.

## 1 Introduction

Generative Adversarial Networks (GANs), particularly StyleGAN2 Karras et al. (2020), have achieved impressive performance in generating photorealistic images. A key area of research, is developing controls for editing these images. The rise of vision-language models (VLMs) like CLIP Radford et al. (2021) has enabled text-guided manipulation of the GAN latent space.

However, existing methods present a dilemma for users, as illustrated in Fig. 1. Optimization-based techniques Patashnik et al. (2021) excel at preserving the subject's identity, resulting in low perceptual error. However, this often leads to overly cautious edits that fail to make significant changes requested by the user. Meanwhile, methods using a single global edit vector can make strong changes but often suffer from attribute entanglement, and in some cases, completely altering identity and background features.

In an attempt to solve this, we introduce DynaVect, a framework that finds the middle ground. Our contribution is a hybrid system that combines the strengths of both approaches. We start with a pre-computed global direction. This starting point provides the power for more significant edits. If we started with something "lighter", tuning down the line will be challenging. Following that, we use a lightweight, trained Dynamic Contextual Modulator to predict a correction vector based on the specific features of the source image. This allows the edit to be strong yet personalized.

Our qualitative results and a large 2AFC study on Amazon Mechanical Turk (**N=3965** pairwise votes show that DynaVect is preferred in **75.6%** of trials overall. More details in §4.6).

**Contributions.** (1) A hybrid editing framework combining a strong global direction with a lightweight modulator for personalization. (2) an identity-preserving orthogonal projection in latent space to reduce entanglement. (3) an optimization-distillation scheme that first runs slow optimization to obtain high-quality edits, then trains a fast feed-forward modulator to approximate the per-image results. (4) comprehensive evaluation highlighting the limitations of LPIPS/L2 for semantic edits.

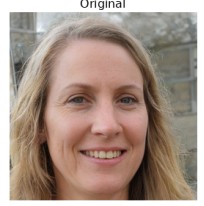 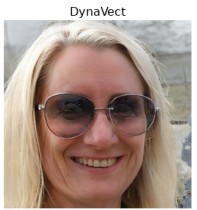 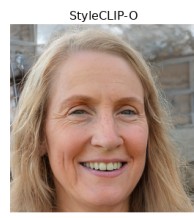 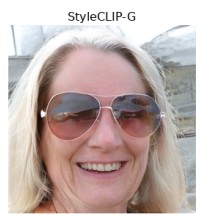 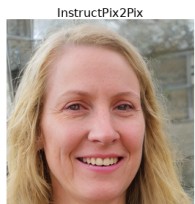 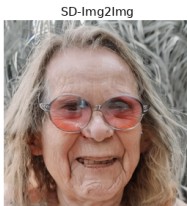

Figure 1: DynaVect successfully performs a complex, multi-attribute edit ("smile + age + hair"). **StyleCLIP-O (Optimizer)** produces an overly subtle edit, failing to make the requested changes. **StyleCLIP-G (Global Direction)** applies the changes but loses the subject's identity. **DynaVect (Ours)** achieves the best balance, implementing all attributes while preserving identity.

**Why hybrid?** Optimizer methods preserve identity but will usually *under-edit* because LPIPS/L2 penalizes change. Meanwhile, single global vectors *over-edit* and entangle attributes. DynaVect starts strong with $\Delta w_{\text{global}}$, personalizes with $\Delta w_{\text{context}}$ predicted from (image, text). It then removes identity aligned components via projection. This balances edit strength with identity preservation.

## 2 Related Work

Our work is situated at the intersection of generative image models and text-guided image manipulation, building upon recent advancements in VLMs.

**Generative Adversarial Networks.** Generative Adversarial Networks (GANs), introduced by Goodfellow et al. Goodfellow et al. (2014), have become the standard for high-fidelity image synthesis. Our work utilizes StyleGAN2 Karras et al. (2020), which refines the architecture of StyleGAN Karras et al. (2019). A key design in the StyleGAN family is the intermediate latent space $\mathcal{W}$, obtained by mapping a noise vector $z \in \mathcal{Z}$ through a learned network. Many editing methods operate in $\mathcal{W}^+$, the per-layer extension of $\mathcal{W}$ that supplies a separate $w$ code to each synthesis layer (e.g., 18 layers for the 1024px generator), enabling fine, layer-wise control over semantics and making the model particularly responsive to editing.

**Text-Guided Image Editing.** The introduction of large-scale VLMs, most notably CLIP Radford et al. (2021), revolutionized text-guided manipulation. CLIP's ability to embed both images and text into a semantically rich space allows for the measurement of text and image alignment via cosine similarity. Changes can now be described in natural language, rather than using a manual process.

**Improving on Context-Agnostic Mappers.** The work most closely related to ours is StyleCLIP Patashnik et al. (2021). This paper demonstrated three distinct methods for using CLIP to guide StyleGAN edits. Our work is an extension of their third and most efficient approach: the latent mapper.

While the original StyleCLIP mapper learns a single global edit vector for a given text prompt, a "one-size-fits-all" approach can struggle. This is because the edit often depends on the source image's unique features. A global vector that successfully adds a smile to one face may fail or cause unwanted artifacts on another. Our work directly addresses this limitation. Instead of applying a static global vector, we introduce a *Dynamic Contextual Modulator* that predicts a personalized correction based on the source image's features. This allows our method to steer the global direction, making the final edit more robust while retaining the subject's identity.

## 3 Method

**Pipeline.** We first form a combined edit $\Delta w_{\text{final}} = \Delta w_{\text{global}} + \alpha \, \Delta w_{\text{context}}$, then remove protected components $\Delta w_{\text{ortho}} = \Delta w_{\text{final}} - \text{Proj}_{\mathcal{B}}(\Delta w_{\text{final}})$, and finally apply $w_{\text{edit}} = w_{\text{source}} + \Delta w_{\text{ortho}}$.

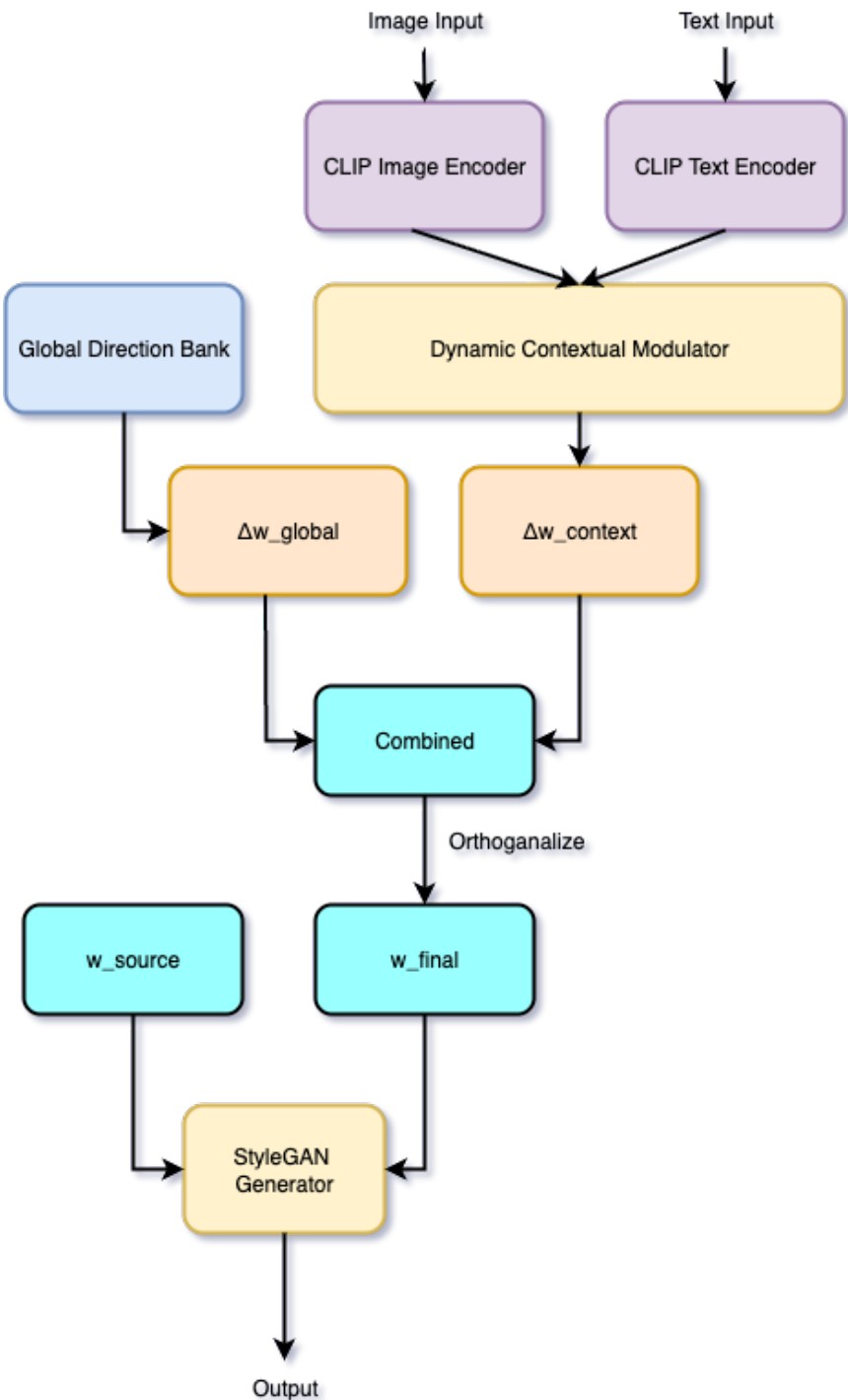

Figure 2: The DynaVect architecture. A source image and target text are encoded by CLIP. These features are fed into our trained **Dynamic Contextual Modulator (DCM)** to predict a context-aware correction, $\Delta w_{\text{context}}$. This is added to a pre-computed **Global Direction**, $\Delta w_{\text{global}}$. The combined vector undergoes **Orthogonalization** (enabled in main results) to reduce drift in protected attributes (identity by default) before being applied to the source latent code, $w_{\text{source}}$, to generate the final edited image.

**Notation recap.** $\Delta w_{\mathrm{global}}$: global direction from sampling; $\Delta w_{\mathrm{context}}$: modulator output; $\alpha$: modulation strength; $\mathcal{B}$: protected directions (identity by default); $\mathrm{Proj}_{\mathcal{B}}(\cdot)$: projection onto $\mathrm{span}(\mathcal{B})$.

Our goal is to perform text-guided edits that follow the prompts. The edits should be visually significant yet preserve the identity of the source image. DynaVect combines (i) a Global Direction Bank for strong baseline edits and (ii) a lightweight Dynamic Contextual Modulator for personalized correction. An orthogonalization step is enabled by default in the main results to reduce drift in protected attributes (identity by default). Additional protected attributes (e.g., gender/age) are off unless stated. We ablate its effect in Appendix B.

### 3.1 Global Direction Baseline

For a given text pair $(T_S, T_T)$, we first compute a strong, generalizable edit direction. We sample a large batch of latent codes $\{w_i\}$ from the StyleGAN $\mathcal{W}^+$ space and score each corresponding image $G(w_i)$ against the source and target prompts using CLIP. The global direction $\Delta w_{\mathrm{global}}$ is then calculated as the difference between the means of the top-scoring latents for each prompt. This generates a baseline vector for the edit, although it may contain entangled features.

### 3.2 Dynamic Contextual Modulator

To personalize the edit, we introduce our modulator network, $M$. This is a MLP that takes two inputs, namely (1) CLIP image features of the source image $E_I(G(w_{\mathrm{source}}))$ and (2) the CLIP text features of the target prompt $E_T(T_T)$. Its purpose is to predict a contextual correction vector, or "delta":

$$\Delta w_{\mathrm{context}} = M(E_I(G(w_{\mathrm{source}})), E_T(T_T)) \tag{1}$$

This allows the correction to be sensitive to the unique characteristics of the input face.

### 3.3 Hybrid Edit Application

At inference time, the final edit direction is a combination of the global baseline and the contextual delta, controlled by a modulation strength parameter $\alpha$:

$$\Delta w_{\mathrm{final}} = \Delta w_{\mathrm{global}} + \alpha \cdot \Delta w_{\mathrm{context}} \tag{2}$$

This hybrid vector is then orthogonalized and applied to $w_{\mathrm{source}}$ to produce the final edit. This removes the identity aligned component of the edit, reducing entanglement (e.g., "smile" without large identity changes).

### 3.4 Identity-projection in latent space (Enabled in main results)

*Why this reduces entanglement.* Leakage is directional: if an edit direction overlaps protected axes (e.g., identity), unintended drift may occur. Subtracting $B(B^\top B)^{-1}B^\top \Delta w_{\mathrm{final}}$ removes those components, retaining the most useful parts of the edit while preventing leakage.

*All operations in this subsection occur in latent space ($\mathcal{W}/\mathcal{W}^+$). We project the **edit vector** to remove identity-aligned components; we do **not** impose any orthogonality constraint on network weights.*

To keep key attributes stable (identity by default), we remove any part of the edit that points along a small set of protected directions $\mathcal{B}$ (identity by default). Concretely,

$$\Delta w_{\mathrm{ortho}} = \Delta w_{\mathrm{final}} - \mathrm{Proj}_{\mathcal{B}}(\Delta w_{\mathrm{final}}).$$

*Subspace form (order-free). When operating in $\mathcal{W}^+$, we flatten per-layer latents so that $v \in \mathbb{R}^{L \cdot d}$ and each $b_j$ has the same shape. Let $B = [b_1, \ldots, b_k]$ (each $b_j$ flattened to the same dimension). Projection onto $\mathrm{span}(B)$ is $\mathrm{Proj}_{\mathcal{B}}(v) = B(B^\top B)^{-1}B^\top v$. We subtract this from $v$ to obtain $\Delta w_{\mathrm{ortho}}$.*

When $B$ is not orthonormal, doing the projections sequentially in different orders will yield different results and generally differs from the subspace projection.

**Defaults used in main tables.** We protect a single direction (identity; $\mathcal{B} = \{b_{\mathrm{id}}\}$) by default to avoid order dependence. Multi-attribute protection (e.g., gender/age) is *disabled* unless explicitly stated. We estimate $b_{\mathrm{id}}$ with the same sampling used for global directions, using the neutral text pair (e.g., "a face" vs "another face").

We evaluate two variants in ablations: (i) disabling identity orthogonalization, and (ii) a "Global+Ortho" variant that handles multi-attribute edits (e.g., "Smile + Age"). First, we apply the initial global direction, and then orthogonalize each subsequent global direction against all previously applied directions. Edit history is not maintained across separate calls, and this variant does not use identity projection. Orthogonalization can slightly change the $\Delta$CLIP/identity trade-off. Therefore, we report both settings and use identity orthogonalization in the main tables.

*Defaults and ablation.* Unless otherwise stated we protect identity only and set the modulation strength to $\alpha$=0.8. Appendix B ablates $\alpha \in \{0.5, 0.8, 1.0, 1.2\}$. We have also extended the protected set to other attributes such as identity, gender, age, showing improved identity and reduced latent drift with a small $\Delta$CLIP trade-off.

### 3.5 Training via Optimization-Distillation

Training the modulator presents a challenge, as there is no "ground truth" for the ideal $\Delta w_{\mathrm{context}}$. Therefore, we propose a self-supervised training scheme. For a given source latent $w_{\mathrm{source}}$ and target text $T_T$, we run an iterative optimization process to find an "ideal" edited latent $w'_{\mathrm{ideal}}$ that minimizes both a joint CLIP and LPIPS identity loss.

We define the ground-truth contextual delta as $\Delta w_{\mathrm{gt}} = w'_{\mathrm{ideal}} - w_{\mathrm{source}}$. The training objective for our modulator $M$ is to minimize the L2 distance between the prediction and this "ground-truth":

$$\mathcal{L} = \|M(\dots) - \Delta w_{\mathrm{gt}}\|_2 \tag{3}$$

This distills the knowledge from the slow optimization into our feed-forward modulator. We also use a reconstruction loss in image space to ensure visual fidelity.

## 4 Experiments

We evaluate two questions: (1) does **DynaVect** better align with user intent than standard baselines, and (2) are common automated metrics (LPIPS/L2) adequate for more significant semantic edits?

**Headline.** In a 2AFC study on Amazon Mechanical Turk (N=3965 pairwise votes after excluding void questions), participants preferred **DynaVect** in **75.6%** of votes overall (Wilson 95% CI [**74.2, 76.9**]). Split by baseline: **67.8%** vs. StyleCLIP-G (95% CI [**65.2, 70.3**]; $N$=1300), **76.4%** vs. StyleCLIP-O (95% CI [**73.5, 79.2**]; $N$=845), **82.2%** vs. InstructPix2Pix (95% CI [**79.5, 84.7**]; $N$=845), and **79.4%** vs. SD-img2img (95% CI [**76.7, 81.8**]; $N$=975).

### 4.1 Experimental Setup

**Reproducibility details.** We use StyleGAN2-FFHQ and CLIP ViT-B/32 across all experiments. **Global-direction sampling:** 2,000 latents, top-$k$=20 per prompt, batch size 20, truncation $\psi$=0.7. **StyleCLIP (Optimizer):** 100 steps (Adam, lr=$10^{-2}$) with loss weights CLIP:1.0, LPIPS:0.8, latent-L2:0.5. **DynaVect training:** 2,000 steps (Adam, lr=$10^{-4}$) via optimization-distillation using the same fixed loss weights. **Seeds:** faces (10), cars (10), churches (5) as listed in table captions; full seed lists are in the repository. **Protected set $\mathcal{B}$:** identity-only by default (gender/age disabled unless stated).

**Ablation axes.** We sweep $\alpha \in \{0.5, 0.8, 1.0, 1.2\}$ and protect {identity} vs. {identity, gender} vs. {identity, gender, age}. We also evaluate two baselines used only in ablations: *GlobalOnly* (sum of global directions) and *Global+Ortho* (orthogonalization of global directions and no identity projection).

*Note:* Global+Ortho does not use identity/gender/age projection, so protected-set settings will be **N/A** for this baseline.

**Orthogonalization:** Identity orthogonalization is enabled for DynaVect outputs in the main results while baselines are unchanged. **Prompt parity:** latent-space methods use (neutral,target) directional prompts; diffusion editors receive an instruction derived from the same pair with tied random seeds and Euler–Ancestral scheduling by default. **Artifacts:** we release code, checkpoints, and run metadata files including `run_meta.json`, `user_votes.csv`, `survey_metrics_from_pairs.csv`, `correlation_user_votes.csv`, `correlation_user_votes_by_baseline.csv`, and the HTML survey bundle.

## 4.2 Reproducibility

We release a self-contained artifact with: (i) the faces training pairs (`artifact/configs/train_edits_faces.json`) (ii) training metadata (`artifact/training/train_meta.json`) containing steps, optimizer/scheduler, loss weights, seed (if available), and checkpoint SHA256; (iii) the evaluation script and default cohorts for faces/cars/churches; (iv) all CSVs produced by our runner (`benchmark_runs_detailed.csv`, `benchmark_summary.csv`, `significance_wilcoxon.csv`, `correlation_user_votes.csv`).

For non-face domains, the modulator is used *zero-shot* (no retraining). Environment lockfiles and installation instructions are included to ensure reproducibility of metrics and figures.

**Training data and hyperparameters (faces modulator).** We train the Dynamic Contextual Modulator only on faces (StyleGAN2–FFHQ). We use **K = 24** text edit pairs (the exact list is released as `artifact/configs/train_edits_faces.json`) and run **2000** optimization–distillation steps. Optimizer is Adam (lr = $10^{-4}$, weight decay = $10^{-5}$) with a cosine annealing scheduler (T_max = 2000, $\eta_{\min} = 10^{-6}$). Losses: direction MSE (weight 1.0), LPIPS (alex) reconstruction (0.5), and a small latent $\ell_2$ regularizer on the predicted delta (0.01). We use StyleGAN2 with truncation $\psi = 0.7$, noise mode `const`, editing in $\mathcal{W}^+$, and CLIP ViT-B/32 for encodings. Identity orthogonalization is *disabled during training* and used only at inference in the main tables (ablation in App. B). Implementation seed(s) and checkpoint hash are recorded in the artifact (`artifact/training/train_meta.json`).

**Baselines.** We compare against both latent-space and diffusion editors:

- **StyleCLIP (Optimizer)** Patashnik et al. (2021): iterative latent optimization minimizing CLIP + LPIPS + latent-L2.

- **StyleCLIP (Global Direction)** Patashnik et al. (2021): a feed-forward application of a pre-computed edit vector.

- **InstructPix2Pix (IP2P)** Brooks et al. (2023): diffusion-based image-to-image editing via natural-language instructions.

- **Stable Diffusion img2img (SD-i2i)** Rombach et al. (2022): pixel-space editing with matched textual intent.

**Protocol parity.** For every case we fix the *same source image* (identical seed or inversion) across all methods. Latent-space methods use the (neutral,target) pair to compute the directional prompt; diffusion editors receive a matched natural-language instruction derived from the same pair. Random seeds are tied across methods; diffusion uses Euler–Ancestral scheduling with default guidance unless otherwise stated.

**Diffusion editors (recent).** To broaden context beyond StyleCLIP, we include two modern diffusion-based editors: (i) **InstructPix2Pix** (IP2P) Brooks et al. (2023) and (ii) a **Stable Diffusion img2img** baseline. For fairness, we keep the *same source images* across all methods. For diffusion, we run image-to-image editing using the *same text instruction* derived from the latent-space edit prompt (Sec. 3); this ensures the prompt semantics are matched while allowing pixel-space editors to operate on the identical input image. Hyperparameters follow the official defaults with Euler Ancestral scheduler; seed is tied to the GAN seed for reproducibility.

**Metrics and protocol.** Our **primary** automated metric is **Directional CLIP** ($\Delta$CLIP; $\uparrow$), defined as $\Delta\text{CLIP} = s(I_{\text{out}}, T_{\text{target}}) - s(I_{\text{out}}, T_{\text{neutral}})$, where $s(\cdot, \cdot)$ is CLIP cosine similarity and $T_{\text{neutral}}$ is a domain-matched neutral text (e.g., "a face" for FFHQ, "a car" for cars, "a church" for churches). We also report **ArcFace identity** cosine ($\uparrow$) when reliable (non-occluding edits) and its **detection rate**. For completeness we include **LPIPS (alex)** and **L2** (both $\downarrow$), noting they reward minimal change. Finally, we report $\|\Delta W\|$ ($\downarrow$) as a latent-drift/entanglement proxy.

**Statistical tests and metric validity.** Do automated metrics track human judgments? For each case–baseline pair we compute *signed* metric deltas and correlate them with 2AFC vote margins. For higher-is-better metrics (directional CLIP, absolute CLIP, ArcFace, PSNR) we use $\Delta m = m_{\text{ours}} - m_{\text{base}}$; for lower-is-better (LPIPS, L2, $\|\Delta W\|$) we flip the sign $\Delta m = -(m_{\text{ours}} - m_{\text{base}})$ so positive always means "ours improved." Human preference is centered as $v = \text{vote\_share}_{\text{ours}} - 0.5$. We report Spearman's $\rho$ and Kendall's $\tau$ with exact *p*-values. On the combined set ($n$=67 case–baseline pairs), **directional CLIP** shows a weak–moderate positive association with user votes (Spearman $\rho$=0.264, $p$=0.0309; Kendall $\tau$=0.186, $p$=0.0281). Absolute CLIP is weaker and marginal (Spearman $\rho$=0.216, $p$=0.0787; Kendall $\tau$=0.153, $p$=0.0710). LPIPS vs. the original does not track user choice (Spearman $\rho$=0.017, $p$=0.893; Kendall $\tau$=0.003, $p$=0.970). Accordingly, we treat user preference as the primary metric and use $\Delta$CLIP as a secondary proxy.

*Conclusion.* Among automated scores, only directional CLIP ($\Delta$CLIP) shows a statistically significant—albeit modest—association with human judgments; absolute CLIP is marginal and LPIPS vs. the original is uninformative. Accordingly, we treat user preference as the *primary* metric and use $\Delta$CLIP as a secondary proxy, reporting LPIPS/L2 only for completeness.

### 4.3 Real-Image Editing (e4e+PTI, $n$=100)

We evaluate on 100 CelebA-HQ faces inverted with e4e and PTI (mediapipe alignment). Per image we run three non-occluding edits (`e1: smile`, `e2: age+`, `e3: blonde`) and compare DynaVect, StyleCLIP-G, and StyleCLIP-O. We report per-edit *means* over the 100 images for (i) $\Delta$CLIP $\uparrow$, (ii) LPIPS $\downarrow$, (iii) L2 $\downarrow$, and (iv) $\|\Delta W\|$ $\downarrow$. ArcFace identity is reliable on these edits and is reported when detected (we also show detection rate in the aggregate table).

| Method | Edit | $\Delta$CLIP $\uparrow$ | LPIPS $\downarrow$ | L2 $\downarrow$ | $\|\Delta W\|$ $\downarrow$ | Identity $\uparrow$ |
|---|---|---|---|---|---|---|
| **DynaVect (Ours)** | e1 (smile) | -0.0045 | 0.2482 | 0.0450 | 17.64 | – |
| | e2 (age+) | 0.0180 | 0.4163 | 0.1565 | 23.39 | 0.1180 |
| | e3 (blonde) | 0.0213 | 0.2978 | 0.0994 | 19.95 | 0.1425 |
| StyleCLIP-G | e1 (smile) | -0.0078 | 0.6796 | 0.4877 | 180.12 | 0.1139 |
| | e2 (age+) | 0.0002 | 0.8423 | 0.8225 | 243.27 | – |
| | e3 (blonde) | -0.0015 | 0.8204 | 0.8867 | 208.02 | – |
| StyleCLIP-O | e1 (smile) | 0.0113 | 0.2000 | 0.0264 | 10.35 | 0.2481 |
| | e2 (age+) | 0.0560 | 0.2052 | 0.0274 | 12.25 | – |
| | e3 (blonde) | 0.0545 | 0.2047 | 0.0273 | 11.79 | 0.1946 |

Table 1: **Real images (e4e+PTI, CelebA-HQ, $n$=100).** Means per edit. Identity is ArcFace cosine when detected (blank if missing). DynaVect achieves positive $\Delta$CLIP on e2/e3 with far lower latent drift than StyleCLIP-G ($\|\Delta W\|$). StyleCLIP-O attains low LPIPS/L2 by under-editing.

**Faces and beyond** We do *not* retrain the modulator outside faces. For cars and churches we apply DynaVect zero-shot with domain-appropriate neutral text ("a car", "a church") and shared seeds across all methods. Identity metrics are omitted as not applicable; we report $\Delta$CLIP, PSNR, LPIPS, and attribute success. Full per-case tables are in App. E. Below we show the detailed face-domain comparison (means $\pm$ std over seeds).

| Method | $\Delta$CLIP $\uparrow$ | $\pm$ | Identity $\uparrow$ | $\pm$ | Detect % | LPIPS $\downarrow$ | L2 $\downarrow$ | $\|\Delta W\|$ $\downarrow$ |
|---|---|---|---|---|---|---|---|---|
| **DynaVect (Ours)** | 0.0116 | 0.0205 | 0.1303 | 0.0173 | 0.67 | 0.3208 | 0.1003 | 20.33 |
| StyleCLIP-G | -0.0030 | 0.0154 | 0.1139 | – | 0.33 | 0.7808 | 0.7323 | 210.47 |
| StyleCLIP-O | 0.0406 | 0.0258 | 0.2214 | 0.0378 | 0.67 | 0.2033 | 0.0270 | 11.46 |

Table 2: **Aggregate over e1/e2/e3 (means $\pm$ std).** Primary metric is $\Delta$CLIP (alignment). DynaVect trades some LPIPS/L2 vs. StyleCLIP-O but stays vastly more stable than StyleCLIP-G ($\|\Delta W\|$), with positive alignment.

| Method | Edit | $\Delta$CLIP$\uparrow$ | Identity$\uparrow$ | LPIPS$\downarrow$ | L2$\downarrow$ | $\|\Delta W\|$ $\downarrow$ |
|---|---|---|---|---|---|---|
| **DynaVect** | Age+ | $0.0145 \pm 0.0070$ | $0.303 \pm 0.073$ | $0.479 \pm 0.052$ | $0.1652 \pm 0.0399$ | $15.3566 \pm 0.0000$ |
| **DynaVect** | Blonde | $0.0418 \pm 0.0167$ | $0.636 \pm 0.120$ | $0.329 \pm 0.067$ | $0.0940 \pm 0.0325$ | $11.5628 \pm 0.0000$ |
| **DynaVect** | Smile | $0.0312 \pm 0.0020$ | $0.581 \pm 0.089$ | $0.326 \pm 0.064$ | $0.0785 \pm 0.0232$ | $11.0589 \pm 0.0000$ |
| StyleCLIP-G | Age+ | $0.0052 \pm 0.0135$ | $0.483 \pm 0.077$ | $0.324 \pm 0.042$ | $0.0756 \pm 0.0113$ | $9.7282 \pm 0.0000$ |
| StyleCLIP-G | Blonde | $0.0573 \pm 0.0177$ | $0.427 \pm 0.140$ | $0.431 \pm 0.067$ | $0.1837 \pm 0.0571$ | $15.2564 \pm 0.0000$ |
| StyleCLIP-G | Smile | $0.0310 \pm 0.0032$ | $0.611 \pm 0.100$ | $0.344 \pm 0.057$ | $0.0999 \pm 0.0376$ | $11.4325 \pm 0.0000$ |
| StyleCLIP-O | Age+ | $0.0570 \pm 0.0267$ | $0.837 \pm 0.039$ | $0.026 \pm 0.005$ | $0.0023 \pm 0.0008$ | $9.6013 \pm 1.2799$ |
| StyleCLIP-O | Blonde | $0.1218 \pm 0.0224$ | $0.828 \pm 0.058$ | $0.030 \pm 0.007$ | $0.0062 \pm 0.0030$ | $9.9427 \pm 0.7271$ |
| StyleCLIP-O | Smile | $0.0379 \pm 0.0079$ | $0.889 \pm 0.036$ | $0.021 \pm 0.005$ | $0.0019 \pm 0.0005$ | $8.1900 \pm 0.8977$ |

Table 3: **Faces (FFHQ).** Means $\pm$ std over seeds. Identity is ArcFace cosine when detected (all three non-occluding edits). Full cars/churches tables are in App. E.

## 4.4 Quantitative Comparison

We compare **DynaVect** with StyleCLIP's global-direction (**StyleCLIP-G**) and optimizer (**StyleCLIP-O**). Table 4 covers non-occluding edits where ArcFace is reliable; Table 5 covers an occluding edit (sunglasses) where identity is omitted.

**Real-image summary.** On e4e+PTI inversions (Tables 1, 2), **DynaVect** achieves positive $\Delta$CLIP on stronger edits (e2/e3) while keeping latent drift *an order of magnitude lower* than StyleCLIP-G ($\|\Delta W\|$ 20.3 vs. 210.5). StyleCLIP-O reaches the best LPIPS/L2 by under-editing; our user study nonetheless prefers DynaVect, indicating better prompt following with reasonable preservation.

**DynaVect** improves identity over StyleCLIP-G on *Smile* and *Age+*, and attains higher $\Delta$CLIP on multi-attribute (non-occluding) edits with a slight identity trade-off. StyleCLIP-O achieves the best identity/LPIPS by minimally editing, inflating similarity metrics.

*Note.* LPIPS/L2 reward minimal change; we report them for completeness but emphasize identity and user preference for semantic edits.

### 4.4.1 Results Summary

- **Ablation recap.** Appendix B shows that enlarging the protected set (identity→identity+gender+age) increases identity and reduces latent drift at a small $\Delta$CLIP cost; within-call orthogonalization of global directions alone does not provide these gains.

- **Versus StyleCLIP-G (feed-forward baseline).** On non-occluding edits, DynaVect improves identity on *Smile* and *Age+* while remaining competitive on $\Delta$CLIP; on the multi-attribute *Combo-NonOcc* task, DynaVect attains the highest $\Delta$CLIP among feed-forward methods again with a modest identity trade-off. For *Blonde*, StyleCLIP-G retains higher identity/LPIPS.

- **Versus StyleCLIP-O (optimizer baseline).** While StyleCLIP-O attains the lowest LPIPS/L2 by under-editing, participants still preferred **DynaVect** in **76.4%** of votes vs. StyleCLIP-O and

| Method | Task | ΔCLIP ↑ | Identity ↑ | LPIPS ↓ | L2 ↓ |
|--------|------|---------|------------|---------|------|
| DynaVect | Smile | 0.0247 ± 0.0052 | **0.672** ± 0.067 | 0.303 ± 0.048 | 0.0743 ± 0.0276 |
| StyleCLIP-G | Smile | **0.0316** ± 0.0025 | 0.519 ± 0.106 | 0.367 ± 0.066 | 0.0991 ± 0.0289 |
| StyleCLIP-O | Smile | 0.0380 ± 0.0074 | 0.892 ± 0.038 | 0.020 ± 0.004 | 0.0019 ± 0.0006 |
| DynaVect | Age+ | 0.0105 ± 0.0076 | **0.372** ± 0.105 | 0.402 ± 0.050 | 0.1059 ± 0.0234 |
| StyleCLIP-G | Age+ | 0.0103 ± 0.0114 | 0.277 ± 0.062 | 0.495 ± 0.037 | 0.1690 ± 0.0708 |
| StyleCLIP-O | Age+ | **0.0568** ± 0.0264 | 0.836 ± 0.044 | 0.026 ± 0.006 | 0.0023 ± 0.0008 |
| DynaVect | Blonde | 0.0545 ± 0.0188 | 0.461 ± 0.144 | 0.437 ± 0.071 | 0.2010 ± 0.0597 |
| StyleCLIP-G | Blonde | 0.0534 ± 0.0219 | **0.596** ± 0.113 | **0.305** ± 0.052 | **0.0919** ± 0.0337 |
| StyleCLIP-O | Blonde | **0.1225** ± 0.0230 | 0.824 ± 0.056 | 0.031 ± 0.007 | 0.0063 ± 0.0030 |
| DynaVect | Combo-NonOcc | **0.0659** ± 0.0114 | 0.250 ± 0.084 | 0.507 ± 0.048 | 0.2106 ± 0.0466 |
| StyleCLIP-G | Combo-NonOcc | 0.0596 ± 0.0192 | **0.297** ± 0.082 | **0.480** ± 0.048 | **0.2196** ± 0.0594 |
| StyleCLIP-O | Combo-NonOcc | 0.1066 ± 0.0332 | 0.811 ± 0.047 | 0.032 ± 0.005 | 0.0047 ± 0.0012 |

Table 4: Non-occluding edits. **Bold** = best among feed-forward/global methods (ours vs. StyleCLIP-G); underline = optimizer (best overall but under-edits). DynaVect improves identity vs. StyleCLIP-G on Smile and Age+, and achieves higher ΔCLIP on Combo-NonOcc with a modest identity trade-off. ΔCLIP denotes the *directional* CLIP score (target vs. neutral "a face") as defined in the Metrics paragraph.

| Method | Task | ΔCLIP ↑ | LPIPS ↓ | L2 ↓ |
|--------|------|---------|---------|------|
| DynaVect | Combo-Occ | 0.1050 ± 0.0179 | 0.558 ± 0.042 | 0.2036 ± 0.0730 |
| StyleCLIP-G | Combo-Occ | 0.1020 ± 0.0176 | 0.609 ± 0.046 | 0.2850 ± 0.0956 |
| StyleCLIP-O | Combo-Occ | 0.1211 ± 0.0321 | 0.033 ± 0.003 | 0.0037 ± 0.0009 |

Table 5: Occluding edit (sunglasses). ArcFace identity is unreliable due to occlusion and is omitted.

**67.8%** vs. StyleCLIP-G (95% CIs reported below), indicating better alignment with user intent despite less favorable similarity metrics.

- **Orthogonalization ablation.** In the main tables, we enable *identity-only* orthogonalization *for DynaVect outputs* (StyleCLIP baselines are unchanged). In ablations we (i) disable orthogonalization for DynaVect and (ii) evaluate a "Global+Ortho" variant that orthogonalizes each newly added *global* direction against the directions already applied *within the same multi-attribute call*. Identity orthogonalization can modestly shift the ΔCLIP/identity trade-off (App. B).

- **Overall.** Automated scores are inconsistent proxies for edit success. LPIPS/L2 reward minimal changes, and ΔCLIP can be high even for outputs with almost no visible changes. The results support DynaVect as a better balance of edit strength and identity preservation, compared to a static global direction.

**Takeaways.** Trends match our qualitative grids and user study. No single automated metric captures edit success fully. LPIPS/L2 reward minimal change, while ΔCLIP can be high even for edits that appear subtle to humans. Therefore, we report both ΔCLIP and ArcFace (when applicable) but rely mainly on user preferences as the primary measure of success.

**Ablation (summary).** "Global+Ortho" slightly improves ΔCLIP but hurts identity, while DynaVect yields the best identity at competitive ΔCLIP. Full results are in Appendix B.

As shown in Tables 4 and 5, a somewhat naive interpretation of the metrics would suggest the Optimizer baseline is the best. In the studies, it achieves the lowest perceptual and pixel-level change (LPIPS, L2). However, this quantitative success is misleading and reveals a fundamental paradox in using such metrics for semantic editing. LPIPS and L2 are designed to measure similarity to a source image, operating on the assumption that the source is the ground truth. While this is valid for tasks like image restoration,

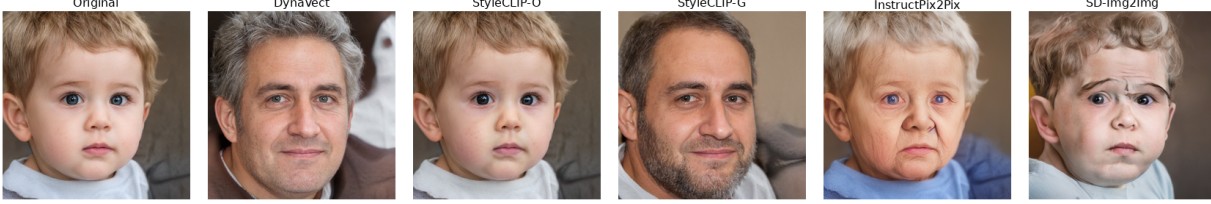

Figure 3: Qualitative comparison on two different subjects. In both cases, the Optimizer baseline (StyleCLIP-O) produces minimal changes for complex edits like aging or changing hair color. The Global Direction baseline (StyleCLIP-G) applies the edits but damages the subject's identity. Our method, DynaVect, successfully implements the attributes while preserving the original identity.

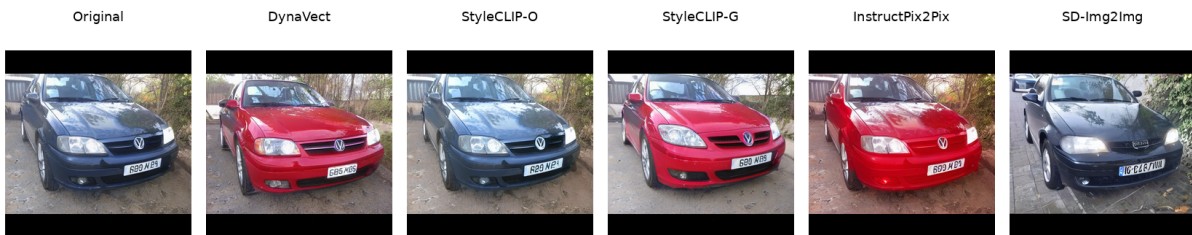

Figure 4: Qualitative comparison for cars. In both cases, the Optimizer baseline (StyleCLIP-O) produces minimal changes for complex edits like aging or changing hair color. The Global Direction baseline (StyleCLIP-G) applies the edits but damages the subject's identity. Our method, DynaVect, successfully implements the attributes while preserving the original identity.

it contradicts the goal of editing, which is to create a meaningful difference. For significant semantic edits like "add age," a large perceptual change is not only expected but *required*. The Optimizer, when heavily penalized by a metric that punishes change, "cheats" the edit's intent by making almost no visible changes at all. This allows it to achieve an almost perfect score on the metric while completely failing the actual goal. This highlights a limitation in standard metrics and underscores the necessity of the qualitative and user-based evaluation. We also observe cases where StyleCLIP-O attains high $\Delta$CLIP despite minimal perceived change. This indicates that $\Delta$CLIP alone can overestimate semantic success when visual salience is low. Therefore, this motivates our use of user studies.

## 4.5 Qualitative Results

Qualitative evaluation reveals the true performance of each method. As shown in the examples in Fig. 4, the Optimizer baseline barely makes any changes, resulting in an image that is nearly identical to the original. The Global Direction baseline successfully changes attributes but at times degrade the subject's identity. Our method, DynaVect, successfully implements changes while maintaining a higher degree of fidelity to the original subject's core features. This demonstrates a better balance that is not captured by traditional metrics.

## 4.6 User Study

We ran a two-alternative forced choice (2AFC) study on Amazon Mechanical Turk. Each trial showed a prompt and two images (ours vs. a baseline) and asked: "Which image better matches the text prompt?" We excluded a small set of void questions with missing images (q7, q8, q14, q15, q23, q27, q28, q52, q69) from analysis.

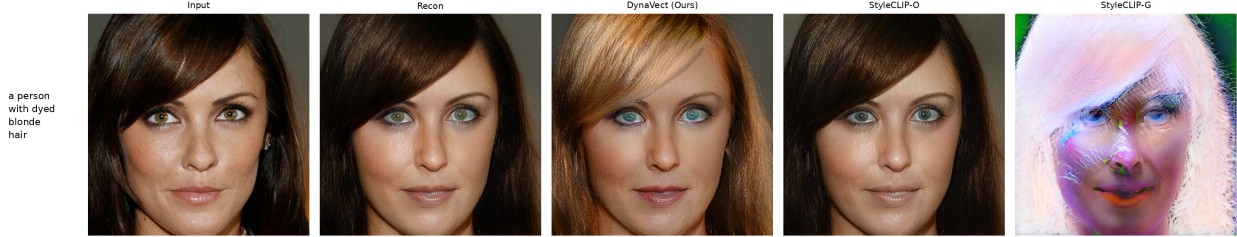

Figure 5: **Real-image edits (e4e+PTI).** Each row: Input, Recon, DynaVect, StyleCLIP-O, StyleCLIP-G. Edits: A person with blonde hair. DynaVect follows prompts while avoiding the large drift seen in StyleCLIP-G; StyleCLIP-O often under-edits.

Overall, DynaVect was preferred in **75.6%** of votes (Wilson 95% CI [**74.2, 76.9**]; $N$=3965). Split by baseline:

| Comparison | Wins / Total (DynaVect) | Preference (%) | 95% CI (Wilson) |
|---|---|---|---|
| DynaVect vs. InstructPix2Pix | 695 / 845 | 82.2% | [79.5, 84.7] |
| DynaVect vs. SD-img2img | 774 / 975 | 79.4% | [76.7, 81.8] |
| DynaVect vs. StyleCLIP-G | 881 / 1300 | 67.8% | [65.2, 70.3] |
| DynaVect vs. StyleCLIP-O | 646 / 845 | 76.4% | [73.5, 79.2] |

Table 6: 2AFC user preferences for DynaVect vs. each baseline. Overall: 2996 / 3965 = 75.6% (95% CI [74.2, 76.9]).

### 4.7 Training Stability

The training of our Dynamic Contextual Modulator was observed to converge well. Figure 6 shows the progression of the key loss components during our optimization-distillation process. The decrease in both the direction and reconstruction losses indicate that the modulator successfully learned to approximate the target edits.

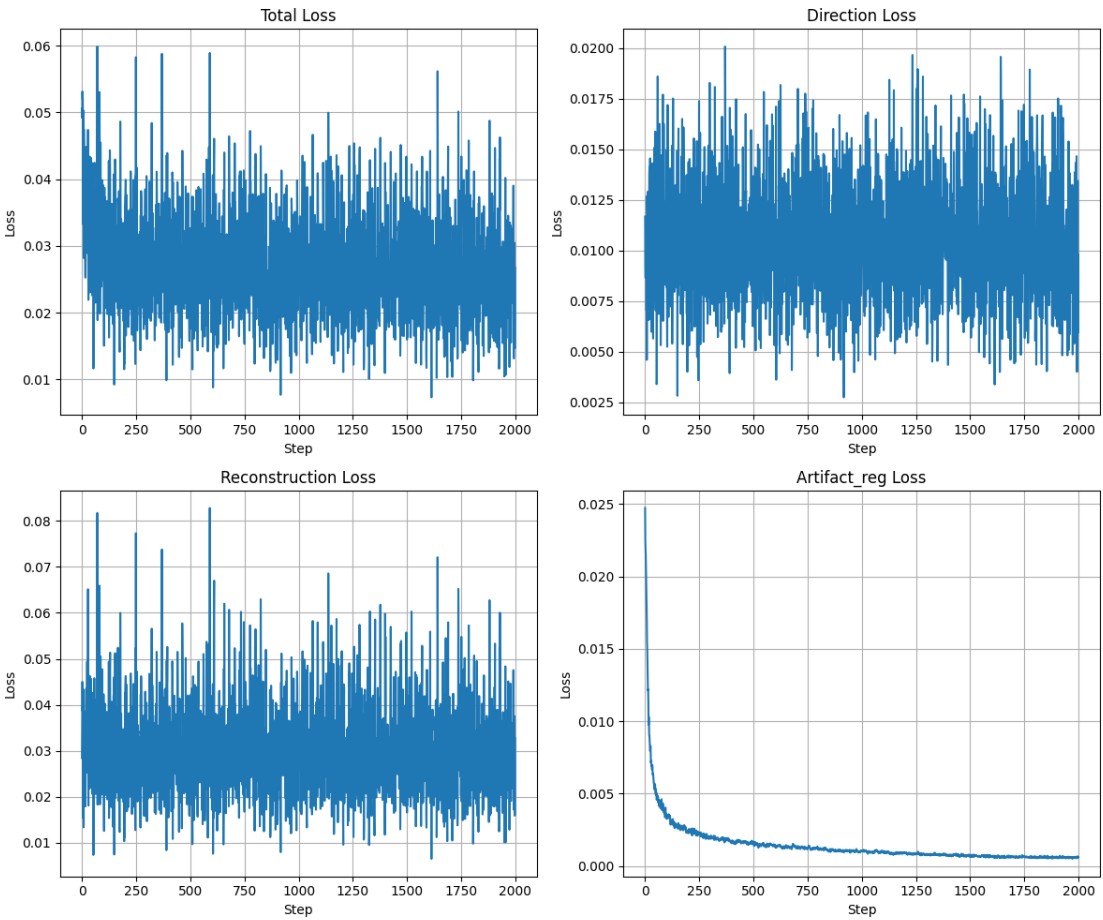

Figure 6: Training loss curves for the Dynamic Contextual Modulator. The model shows stable convergence, with the regularization loss (bottom right) decreasing as expected.

## 5 Conclusion

We have presented DynaVect, a hybrid framework that improves the controllability of text-guided GAN editing. Our key contribution is a lightweight Dynamic Contextual Modulator, which personalizes a global edit vector based on the features of the source image. By training this modulator with an optimization-distillation technique, our method produces significant semantic changes that better align with user intent, while preserving subject identity. We enable identity-orthogonalization in the main results and ablate its effect (including a no-orthogonalization variant) in Appendix B.

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

## A  Modulator Architecture

The detailed architecture of our Dynamic Contextual Modulator is provided in Table 7. The network is a standard Multi-Layer Perceptron (MLP) that processes concatenated CLIP features.

| Layer Type | Parameters |
|---|---|
| Input | 2 x CLIP Features (Total: 1024-dim) |
| Linear | $1024 \rightarrow 2048$ |
| LayerNorm, LeakyReLU, Dropout(0.1) | - |
| Linear | $2048 \rightarrow 2048$ |
| LayerNorm, LeakyReLU, Dropout(0.1) | - |
| Linear | $2048 \rightarrow 1024$ |
| LayerNorm, LeakyReLU | - |
| Layer Heads (x18) | $1024 \rightarrow 512$ (w_dim) |
| Attention Head | $1024 \rightarrow 18$ (num_layers) + Softmax |

Table 7: Architecture of the Dynamic Contextual Modulator.

## B  Ablation Study

### B.1  Ablations: modulation strength and protected sets

**Setup.** We sweep $\alpha \in \{0.5, 0.8, 1.0, 1.2\}$ and progressively enlarge the protected set $\mathcal{B}$ from identity to identity+gender and identity+gender+age. We compare three families: **GlobalOnly** (sum of sampled global directions), **Global+Ortho** (orthogonalize each subsequent global direction to the previously applied ones within the same call; no identity projection), and **DynaVect** (ours; identity-projection enabled). Full per-cell statistics are released in `ablation_summary.csv`; per-seed rows in `ablation_long.csv`.

**Key trends.** (1) For **DynaVect**, enlarging the protected set improves ArcFace identity and reduces $\|\Delta W\|$ with a small decrease in $\Delta$CLIP; curves are flat across $\alpha$, so we set $\alpha{=}0.8$ by default. (2) **Global+Ortho** is insensitive to protected sets (no identity projection; *protected = N/A*) and mainly changes the cosine to the global direction. (3) **GlobalOnly** closely follows the global vector (cos$\approx$ 1), giving higher $\Delta$CLIP but worse identity.

**Reproducibility (command).** `python ablation.py \`
`-edits_json '[{"neutral":"a face","target":"a smiling face","strength":0.9},{"neutral":"a`
`young person","target":"an old person","strength":0.7}]' \`
`-alpha_grid "0.5,0.8,1.0,1.2" \`
`-protected_sets "identity; identity+gender; identity+gender+age" \`
`-out_dir "ablate_all"`

| Method & Prot. | $\alpha$ | $\Delta$CLIP ↑ | Identity ↑ | LPIPS ↓ | $\|\Delta W\|$ ↓ | cos↑ |
|---|---|---|---|---|---|---|
| DynaVect / id | 0.8 | 0.0403 | 0.3947 | 0.4282 | 15.33 | 0.486 |
| DynaVect / id+gender | 0.8 | 0.0371 | 0.4232 | 0.4193 | 14.74 | 0.470 |
| DynaVect / id+gender+age | 0.8 | 0.0370 | 0.4418 | 0.4139 | 14.65 | 0.404 |
| Global+Ortho / (n/a) | 0.8 | 0.0395 | 0.4328 | 0.3690 | 14.29 | 0.965 |
| GlobalOnly / (n/a) | 0.8 | 0.0392 | 0.4663 | 0.3399 | 11.96 | 1.000 |

Table 8: **Ablation snapshot (FFHQ).** Means over seeds at $\alpha{=}0.8$. Enlarging the protected set in DynaVect increases identity and reduces drift ($\|\Delta W\|$), with a small $\Delta$CLIP drop. Global baselines ignore identity projection; *GlobalOnly* tracks the global direction (cos$\approx$1). Full sweeps are in `ablation_summary.csv`.

## C    Additional Qualitative Results

This section provides the remaining qualitative results grids to demonstrate the consistency of our method across various subjects.

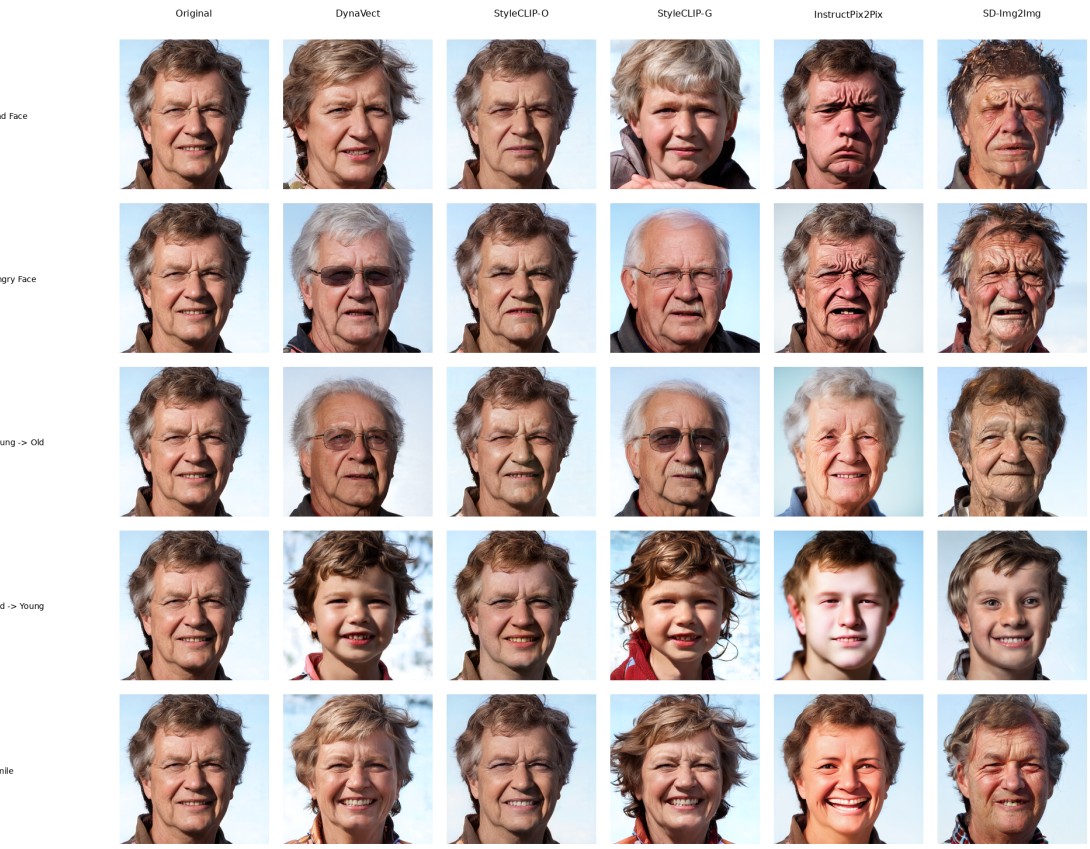

Figure 7: Additional qualitative results for Seed 27 for faces.

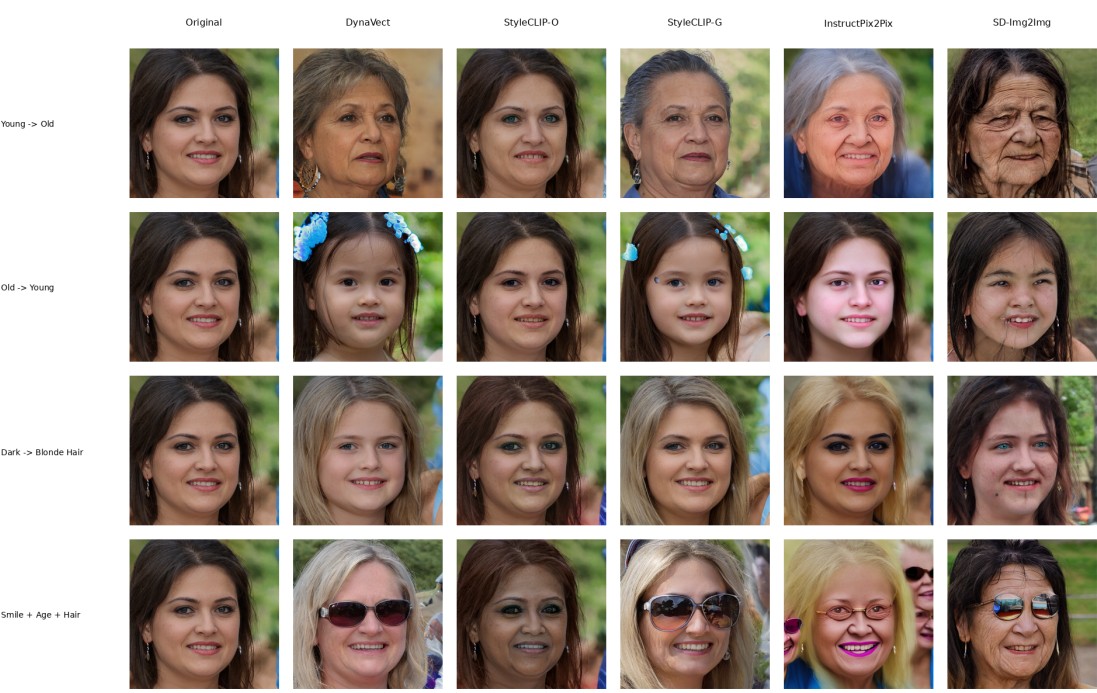

Figure 8: Additional qualitative results for Seed 4126 for faces.

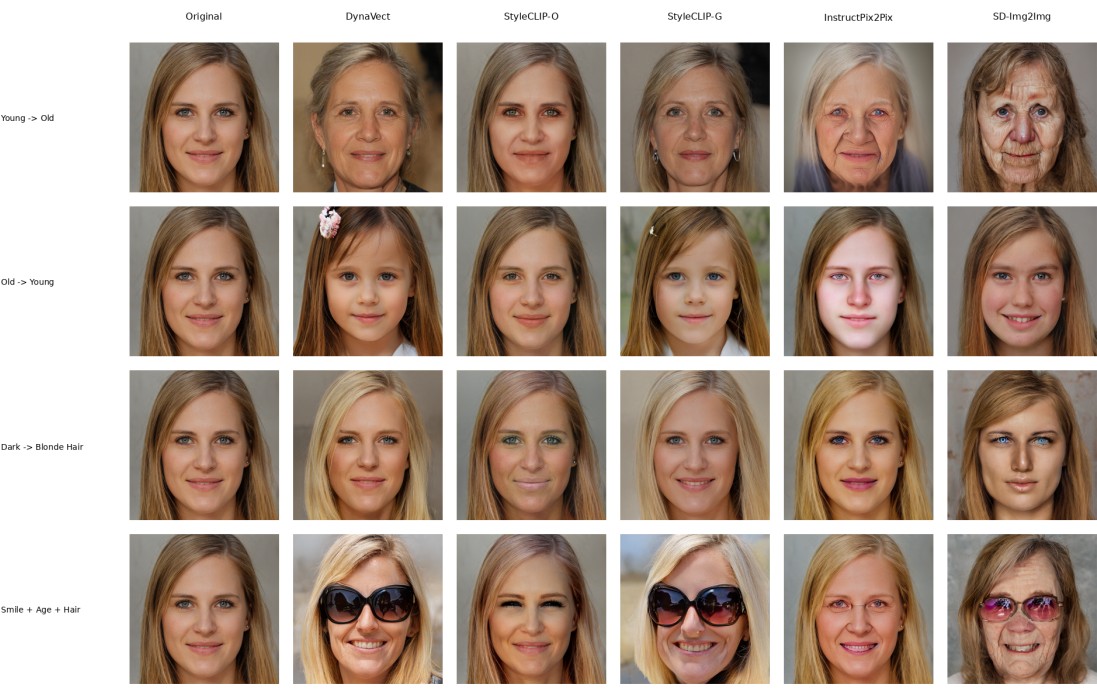

Figure 9: Additional qualitative results for Seed 1776 for faces.

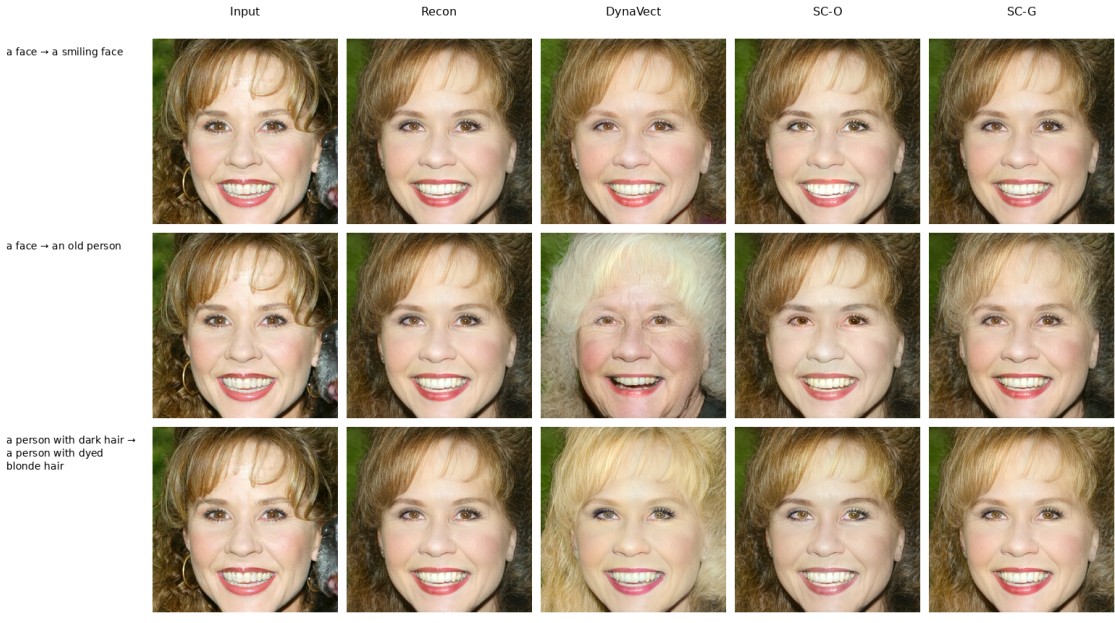

Figure 10: Additional qualitative results for Image 14247 from Celeb-A-HQ dataset for faces.

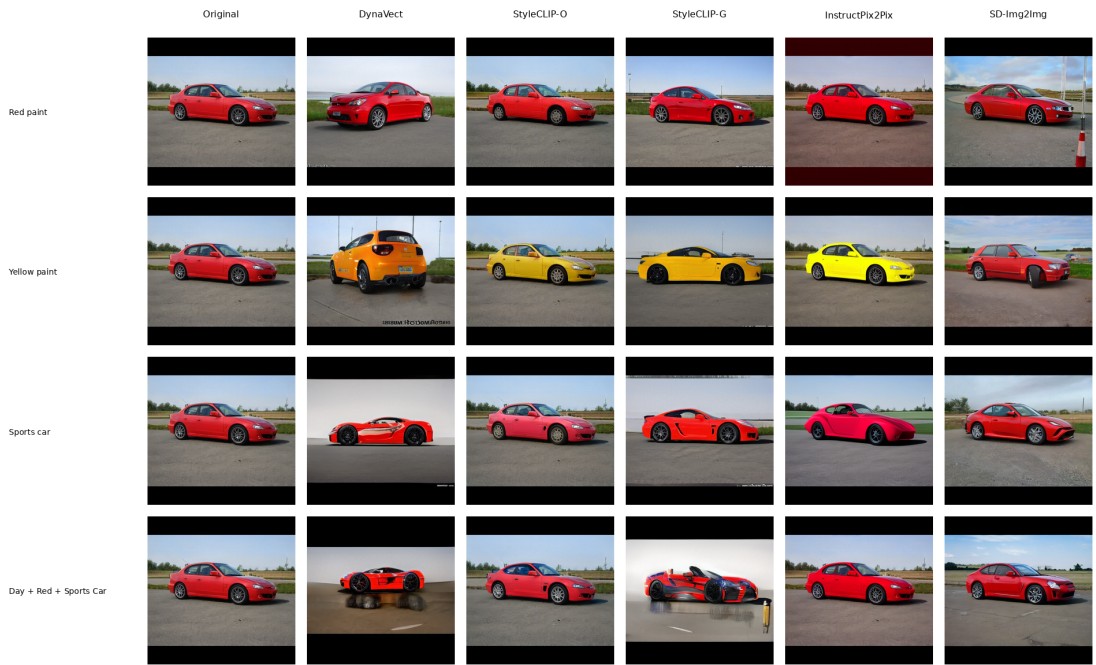

Figure 11: Qualitative results for Seed 58 for cars.

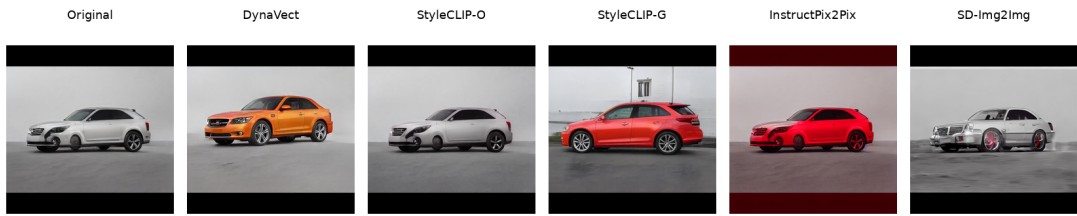

Figure 12: Qualitative results for Seed 37 for cars. Prompt was a sports car

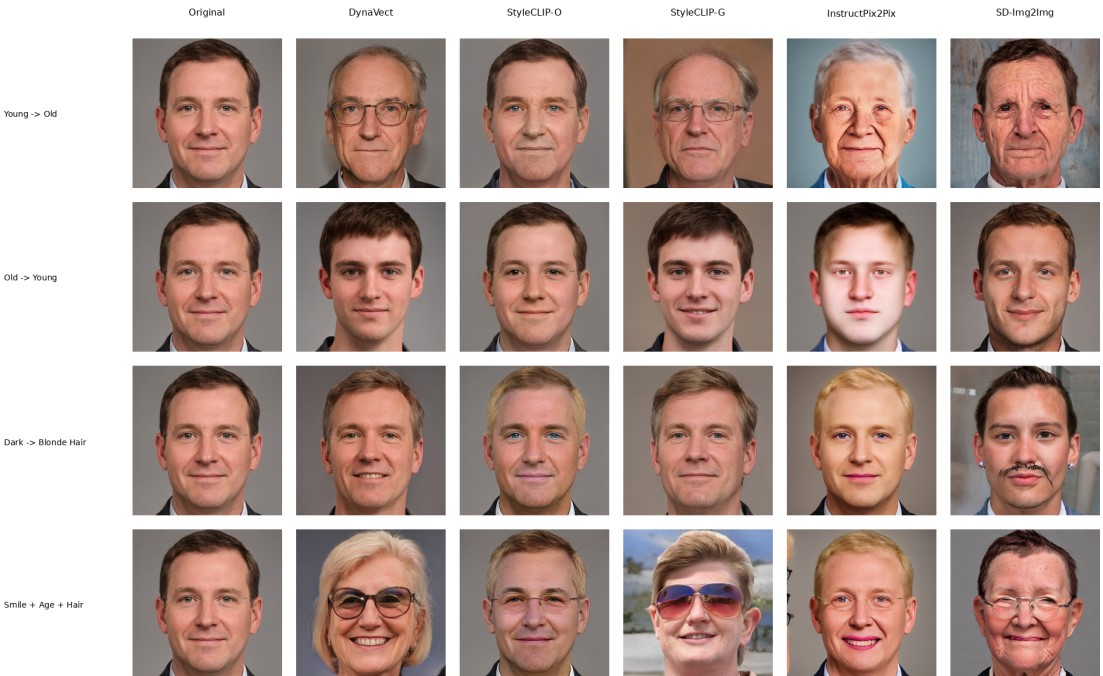

Figure 13: Failure case due to concept entanglement. The gender-neutral prompt "a person with dyed blonde hair" incorrectly feminizes the male subject. Seed 4540 for faces

# D  Limitations and Failure Cases

While DynaVect demonstrates robust performance on these edits, it is still subject to the limitations of the underlying StyleGAN and CLIP models. In this section, we present a qualitative analysis of common failure modes.

## D.1  Dataset Bias and Concept Entanglement

We observe failures that stem from strong dataset biases within the base models, which leads to concept entanglement. For example, the concept of "blonde hair" is heavily correlated with the concept of "woman" in the training data. As a result, applying a gender-neutral prompt like "a person with dyed blonde hair" to a male subject can cause an unwanted shift in gender towards female, as shown in Fig. 13.

## D.2  Limitations in Color Transfer: Red → Yellow

In the cars domain we observe a failure to repaint most red cars to yellow. This stems from (i) the *directional prompts*: using a neutral of "a car" and a target of "a yellow car" drives the edit toward image-level correlations of yellow-car photos (different shape car but still red) rather than a paint-color swap (ii) *StyleGAN priors*: red paint is probably overrepresented and strongly anchored in the generator. We theorise that, CLIP-guided edits prefer easier geometry/lighting shifts that will still raise the "yellow car" score (iii) the modulator is *zero-shot* outside faces, so it inherits these entanglements. In practice, using an explicit *red→yellow* pair (neutral: "a red car", target: "a yellow car"), adding negatives ("without red/maroon"), and restricting the delta to color-dominated layers reduces this failure mode.

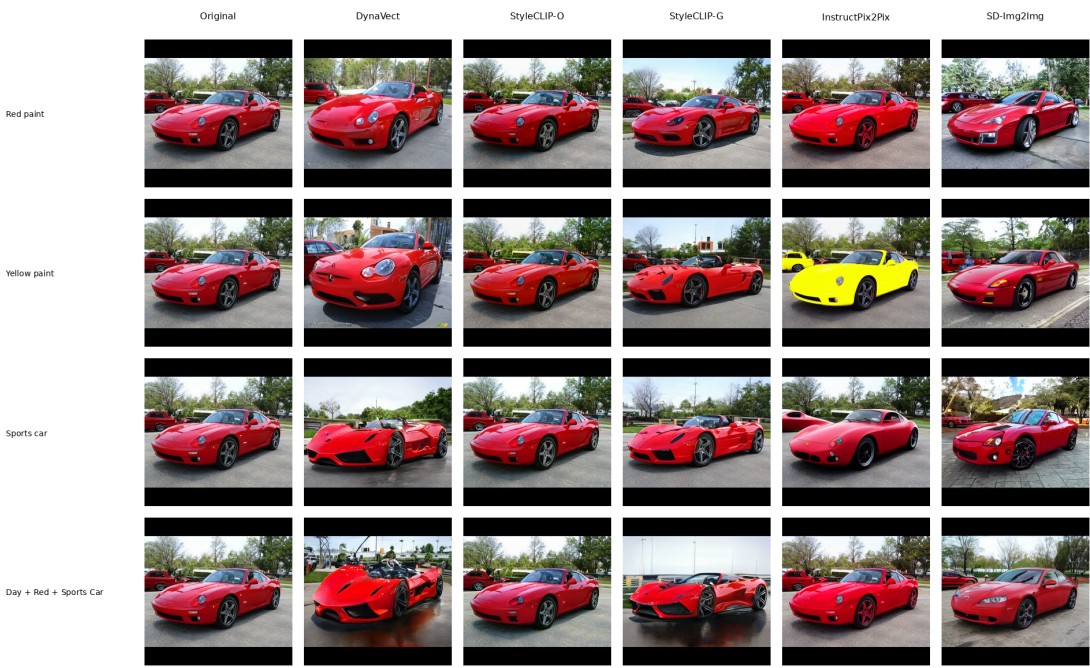

Figure 14: Failure case for color transfer. The prompt was "a yellow car."

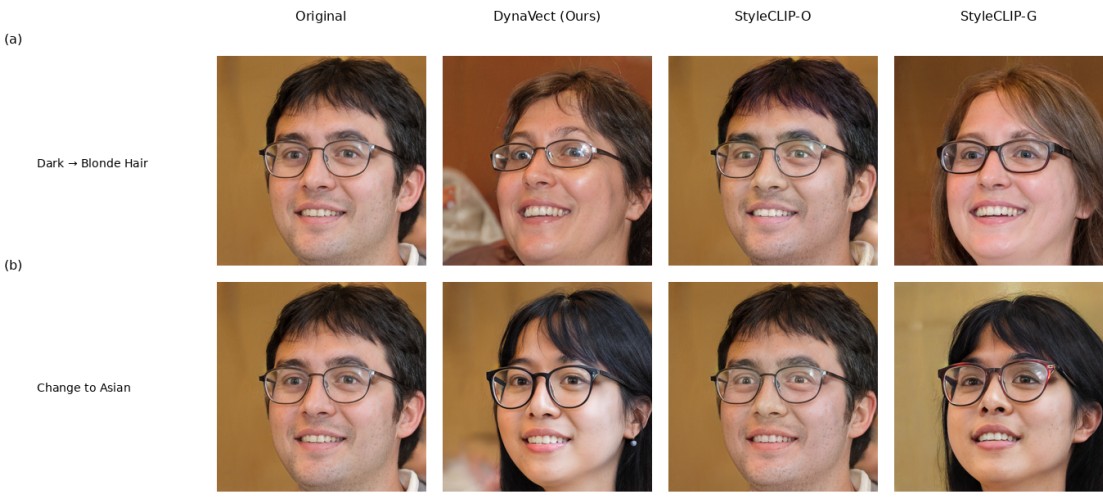

Figure 15: Failure case for a large identity transformation (Subject 2). The prompt was "an Asian face."

# E    Non-face Domains: Cars and Churches

## E.1    Cars (StyleGAN2-Cars)

| Method | Case | $\Delta$CLIP$\uparrow$ | LPIPS$\downarrow$ | L2$\downarrow$ | $\|\Delta W\|\downarrow$ | cos-to-global$\uparrow$ | CLIPabs$\uparrow$ | ContentCos$\uparrow$ | PSNR$\uparrow$ |
|---|---|---|---|---|---|---|---|---|---|
| **DynaVect** | RedPaint | $0.0139 \pm 0.0154$ | $0.354 \pm 0.066$ | $0.1824 \pm 0.0545$ | $22.1601 \pm 0.0000$ | $0.337 \pm 0.288$ | $0.256 \pm 0.021$ | $0.845 \pm 0.069$ | $13.61 \pm 1.42$ |
| **DynaVect** | SportsCar | $0.0192 \pm 0.0164$ | $0.317 \pm 0.056$ | $0.1427 \pm 0.0460$ | $20.6214 \pm 0.0000$ | $0.545 \pm 0.106$ | $0.260 \pm 0.021$ | $0.784 \pm 0.039$ | $14.68 \pm 1.37$ |
| **DynaVect** | YellowPaint | $-0.0011 \pm 0.0292$ | $0.345 \pm 0.065$ | $0.1760 \pm 0.0806$ | $22.2002 \pm 0.0000$ | $0.371 \pm 0.156$ | $0.247 \pm 0.029$ | $0.824 \pm 0.057$ | $13.89 \pm 1.66$ |
| StyleCLIP-G | RedPaint | $0.0133 \pm 0.0148$ | $0.310 \pm 0.056$ | $0.1595 \pm 0.0541$ | $17.9794 \pm 0.0000$ | $0.317 \pm 0.214$ | $0.253 \pm 0.019$ | $0.870 \pm 0.052$ | $14.21 \pm 1.44$ |
| StyleCLIP-G | SportsCar | $0.0201 \pm 0.0139$ | $0.281 \pm 0.041$ | $0.1230 \pm 0.0206$ | $18.5467 \pm 0.0000$ | $0.431 \pm 0.126$ | $0.264 \pm 0.015$ | $0.796 \pm 0.058$ | $15.18 \pm 0.72$ |
| StyleCLIP-G | YellowPaint | $-0.0017 \pm 0.0358$ | $0.400 \pm 0.049$ | $0.2283 \pm 0.0889$ | $25.7440 \pm 0.0000$ | $0.342 \pm 0.130$ | $0.245 \pm 0.041$ | $0.762 \pm 0.047$ | $12.67 \pm 1.41$ |
| StyleCLIP-O | RedPaint | $0.0491 \pm 0.0133$ | $0.030 \pm 0.004$ | $0.0084 \pm 0.0032$ | $10.5956 \pm 1.3873$ | $0.005 \pm 0.016$ | $0.319 \pm 0.015$ | $0.852 \pm 0.036$ | $27.00 \pm 1.38$ |
| StyleCLIP-O | SportsCar | $0.0413 \pm 0.0108$ | $0.026 \pm 0.009$ | $0.0084 \pm 0.0044$ | $9.7514 \pm 1.0156$ | $0.007 \pm 0.014$ | $0.311 \pm 0.012$ | $0.846 \pm 0.058$ | $27.29 \pm 2.08$ |
| StyleCLIP-O | YellowPaint | $0.0663 \pm 0.0194$ | $0.037 \pm 0.005$ | $0.0097 \pm 0.0027$ | $11.1674 \pm 0.9055$ | $0.011 \pm 0.016$ | $0.338 \pm 0.017$ | $0.847 \pm 0.026$ | $26.32 \pm 1.23$ |

Table 9: **Cars.** Means $\pm$ std over seeds. Identity is not applicable

## E.2    Churches (StyleGAN2-Churches)

| Method | Case | $\Delta$CLIP$\uparrow$ | LPIPS$\downarrow$ | L2$\downarrow$ | $\|\Delta W\|\downarrow$ | cos-to-global$\uparrow$ | CLIPabs$\uparrow$ | ContentCos$\uparrow$ | PSNR$\uparrow$ |
|---|---|---|---|---|---|---|---|---|---|
| **DynaVect** | Heavy_Snow | $-0.0025 \pm 0.0203$ | $0.584 \pm 0.108$ | $0.2654 \pm 0.0891$ | $47.9518 \pm 0.0000$ | $0.748 \pm 0.094$ | $0.249 \pm 0.040$ | $0.739 \pm 0.081$ | $11.97 \pm 1.42$ |
| **DynaVect** | Night_LightsOn | $-0.0125 \pm 0.0403$ | $0.645 \pm 0.103$ | $0.5942 \pm 0.3116$ | $60.6176 \pm 0.0000$ | $0.859 \pm 0.034$ | $0.244 \pm 0.037$ | $0.727 \pm 0.131$ | $8.67 \pm 1.93$ |
| StyleCLIP-G | Heavy_Snow | $-0.0227 \pm 0.0253$ | $0.463 \pm 0.067$ | $0.2267 \pm 0.0661$ | $40.8197 \pm 0.0000$ | $0.690 \pm 0.046$ | $0.232 \pm 0.018$ | $0.811 \pm 0.066$ | $12.66 \pm 1.56$ |
| StyleCLIP-G | Night_LightsOn | $0.0019 \pm 0.0380$ | $0.725 \pm 0.088$ | $0.8140 \pm 0.2953$ | $87.0455 \pm 0.0000$ | $0.871 \pm 0.040$ | $0.242 \pm 0.052$ | $0.680 \pm 0.054$ | $7.10 \pm 1.33$ |
| StyleCLIP-O | Heavy_Snow | $0.0616 \pm 0.0085$ | $0.021 \pm 0.003$ | $0.0045 \pm 0.0012$ | $12.6694 \pm 1.2702$ | $-0.008 \pm 0.039$ | $0.371 \pm 0.008$ | $0.803 \pm 0.057$ | $29.64 \pm 1.20$ |
| StyleCLIP-O | Night_LightsOn | $0.0233 \pm 0.0235$ | $0.022 \pm 0.003$ | $0.0058 \pm 0.0032$ | $15.3785 \pm 1.0408$ | $0.005 \pm 0.009$ | $0.322 \pm 0.011$ | $0.820 \pm 0.049$ | $28.91 \pm 2.37$ |

Table 10: **Churches.** Means $\pm$ std over seeds. Identity is not applicable

