# OpenReview forum: "DynaVect: Context-Aware Modulation of Global Edit Direc- tions for Controllable GAN Editing"
_TMLR — Rejected by TMLR_

### Review · Reviewer_LY6N · 2025-09-11

**Summary Of Contributions:**

**Summary**:
The paper proposes a hybrid image-editing framework based GANs, which combines optimizer-based editing methods and the "global direction" editing methods. Comprehensive experiments are conducted. Both quantitative and qualitative results are presented. However, the presentation, the motivation, and the method of the submission are confusing and ambiguous, c.f. "Requested Changes".

**Contributions**:
1. Despite the ambiguous presentation and illustration in Section 3.3, "Orthogonal property" of the weight matrix is still insightful and can further inspire the community of generative models. I guess that he authors want to build a bridge between the "Orthogonal property" and the entanglement of image editing. However, the illustration is very confusing.


2. In terms of "Identity" metric, DynaVect outperforms other baselines.

**Audience:**

Yes

**Audience Explanation:**

The audience of generative models and computer vision will be interested in the topic of image editing. The orthogonal property or the weight matrix is also the trending topic in machine learning.

**Broader Impact Concerns:**

There no ethical implications of the submission.

**Claims And Evidence:**

Yes

**Claims Explanation:**

The submission provides both qualitative and quantitative results, manifesting the superiority of the method.

**Requested Changes:**

The presentation, the method, and the motivation of the submission need major revisions.

1. In section 3.3, the symbols $\Delta w_{\rm ortho}$, $\Delta w_{\rm final}$ and ${\rm Proj}$ are confusing. As a reader, I cannot depict a pipeline of the update rule of the weight $w$.

2. Orthogonal property of the weight matrix is a very important aspect, but the presentation of section 3.3 does not reflect the **relation** between the orthogonal property or the weight matrix and the entanglement of image editing.

3. The motivation is not well illustrated. Why do you combine the optimizer-based methods and the global direction methods? What is the key challenge in image editing? What is the key contribution/insight of your method? Can the combination/hybrid address the challenge in image editing? I cannot find them in the submission.

4. The experiments cannot show the superiority. In Table 1, in terms of "LPIPS", "L2", and "$\Delta {\rm CLIP}$", DynaVect cannot compete with other baselines.

Overall, I sincerely suggest the authors revise the submission and take submissions seriously.

---

### Review · Reviewer_uG7U · 2025-09-24

**Summary Of Contributions:**

The paper outlines the following key contributions:
1. The paper presents a novel approach that integrates a strong global edit direction with a lightweight Dynamic Contextual Modulator (DCM). The DCM personalizes the edit by predicting a correction vector based on CLIP-encoded features of the source image and target text, enhancing control over StyleGAN edits while preserving identity.
2. The paper introduces an innovative technique in the latent space to reduce attribute entanglement. By projecting out components aligned with protected attributes, the method minimizes unintended identity changes during editing.
3. The paper presents a self-supervised training method where a slow optimization process generates high-quality pseudo-ground-truth edits. These are then distilled into a fast feed-forward modulator, approximating per-image optimization results efficiently.

**Audience:**

Yes

**Audience Explanation:**

The paper's focus on improving text-guided GAN editing with a hybrid approach (DynaVect) using StyleGAN2 and CLIP aligns with current trends in vision-language integration and controllable generation, areas of active interest in the ML community. The contributions address practical challenges like attribute entanglement and identity preservation, which are relevant to researchers working on GANs and latent space manipulation. However, the study’s focus on synthetic faces limits its immediate applicability, potentially reducing its appeal to researchers and practitioners interested in broader image editing tasks across diverse domains.

**Claims And Evidence:**

Yes

**Claims Explanation:**

The evidence is accurate (based on established tools like CLIP/StyleGAN2), convincing (supported by user preferences and visuals), and mostly clear (with detailed tables and figures). However, the evaluation is restricted to facial edits on StyleGAN2-FFHQ, which is common but limits claims of broader applicability. No experiments on other domains (e.g., cars, landscapes via StyleGAN-Ada) or models (e.g., diffusion-based like Stable Diffusion). Multi-attribute edits are tested, but complex prompts (e.g., negation like "smile without glasses") or real images (inverted via PTI/e4e) are absent. These gaps slightly temper the strength of support. With expanded experiments, the evidence could fully substantiate the claims.

**Requested Changes:**

Below are my remaining suggestions and questions critical for securing acceptance:
1. Include experiments on non-facial domains (e.g., StyleGAN-Ada for cars/landscapes) and real images (via inversion techniques like PTI or e4e) to test generalization beyond StyleGAN2-FFHQ faces.
2. The ALB is described in detail but disabled in all results. Why include it if not evaluated?
3. Hyperparameters like $\alpha$ (modulation strength) aren't ablated. It's unclear how sensitive results are to it.
4. Training data for DCM isn't quantified. How many pseudo edits?
5. Figure 2 could be restructured with a more professional layout to enhance its clarity and align with the standards of a peer-reviewed publication.

---

> ### Author Response · Authors · 2025-10-08
> **Latest revision**
>
> Latest revisions include non-facial domains and real images. ALB was removed and additional ablation was carried out. Figure 2 was also changed as well as training data for DCM quantified

---

### Review · Reviewer_BfEj · 2025-09-26

**Summary Of Contributions:**

The paper proposes DynaVect, a hybrid method for text-guided image editing in StyleGAN2 using CLIP. It combines a pre-computed global edit direction (for strong changes) with a lightweight Dynamic Contextual Modulator (DCM, an MLP) that predicts a personalized correction vector based on the source image's CLIP features to better preserve identity and reduce attribute entanglement. The final edit vector undergoes orthogonal projection to remove components aligned with protected attributes (e.g., identity). Training uses "optimization-distillation," where pseudo-ground-truth edits are generated via per-image optimization (with fixed CLIP/LPIPS/L2 losses), then distilled into the feed-forward DCM. An optional Adaptive Loss Balancer (ALB) is described but not used in results.

Strengths: Introduces a practical hybrid approach to balance edit strength and identity preservation; uses a user study to demonstrate human preference over baselines; identifies metric shortcomings in editing tasks.

Weaknesses: Relies on dated models (StyleGAN2 from 2020, CLIP ViT-B/32); limited evaluation scope (only faces, small set of prompts, no comparison to recent methods); some components (e.g., ALB) are underexplored or unused; potential reproducibility issues with hyperparameters.

**Audience:**

Yes

**Audience Explanation:**

TMLR's audience includes researchers in machine learning, generative models, and vision-language tasks. The findings on hybrid editing for GANs, personalization via context-aware modulation, and critiques of evaluation metrics (e.g., LPIPS favoring minimal changes) could interest those working on controllable generation or text-guided manipulation, especially in legacy GAN contexts like StyleGAN. The optimization-distillation technique might appeal to distillation/dist Knowledge transfer enthusiasts. However, interest may be limited by the paper's focus on outdated models (StyleGAN2/CLIP from 2020-2021), as the field has shifted toward diffusion models; still, some (e.g., in resource-constrained or face-editing applications) would find the user-study-backed improvements and metric discussions relevant for ongoing debates in semantic editing evaluation.

**Broader Impact Concerns:**

The paper discusses ethical issues like dataset biases leading to entanglement (e.g., "blonde hair" feminizing males) and misuse in identity transformations (e.g., ethnicity changes causing non-photorealism), limiting evaluation to synthetic faces and avoiding biometrics. However, it lacks a dedicated Broader Impact Statement. Concerns include: amplification of biases in CLIP/StyleGAN (trained on uncurated data, potentially perpetuating stereotypes in edits like aging/gender); potential for deepfake-like misuse in facial editing (e.g., non-consensual alterations), not sufficiently mitigated by synthetic-only focus; accessibility issues, as the method requires significant compute for distillation/optimization, excluding low-resource users; environmental impact from training (though lightweight, not quantified). A statement should address these, propose debiasing strategies (e.g., fair prompts), and discuss societal risks like misinformation from edited images.

**Claims And Evidence:**

No

**Claims Explanation:**

The core claims—that DynaVect better balances edit strength and identity preservation, aligns more with user intent (59.6% preference), and reveals metric limitations—are partially supported but lack convincing evidence in several areas. Qualitative figures (e.g., Figs. 1, 3) visually support better identity retention. However, quantitative metrics are inconsistent: DynaVect often underperforms StyleCLIP-O on LPIPS/L2/identity (e.g., Table 1: worse identity on Blonde/Combo-NonOcc) and only modestly improves ΔCLIP in some cases, contradicting claims of superior balance. The evidence for metric limitations is anecdotal (e.g., optimizer "cheats" by under-editing), without rigorous analysis like correlation studies between metrics and user preferences. Orthogonalization's impact is ablated (Table 5), but not deeply analyzed for why it helps/hurts. Training details are vague (e.g., dataset size for distillation not specified, only 2000 steps mentioned), and reproducibility is hindered by undisclosed code/hyperparameters (though promised). No comparisons to post-2021 methods (e.g., recent diffusion-based editors or advanced GAN mappers), weakening novelty claims. Failure cases (e.g., biases in Figs. 9-11) are acknowledged but not quantified.

**Requested Changes:**

Expand comparisons to recent methods (critical): Include baselines like DiffusionCLIP (2022+), InstructPix2Pix, or modern GAN editors (e.g., StyleGAN3 variants). Current baselines are only StyleCLIP (2021), making novelty unclear—reject without broader context.
Quantify metric limitations rigorously (critical): Add statistical analysis (e.g., Spearman correlation between ΔCLIP/LPIPS and user votes) instead of qualitative claims. Without this, claims about metric inadequacies are unconvincing.
Detail training data and reproducibility (critical): Specify dataset size/composition for optimization-distillation, full hyperparameters (e.g., exact loss weights beyond mains, random seeds), and release code early. Vague details hinder verification.
Incorporate/evaluate Adaptive Loss Balancer (strengthening): Since described but disabled, add results with ALB enabled to show if it improves the ΔCLIP/identity trade-off.
Broader ablation studies (strengthening): Ablate modulation strength α, more protected attributes (e.g., gender+age), and test on diverse datasets beyond FFHQ (e.g., CelebA-HQ for real images).
Address biases quantitatively (strengthening): Measure entanglement (e.g., via classifiers for gender/ethnicity shifts) in failure cases, not just qualitatively.
Fix inconsistencies/typos (strengthening): Correct future-dated references (e.g., Guo et al. 2025), clarify orthogonalization variants, and ensure consistent notation (e.g., Δw vs. ∆w).

---

> ### Author Response · Authors · 2025-10-08
> **Latest revision**
>
> 1. "Why no DiffusionCLIP?"
> DiffusionCLIP targets pixel-space diffusion guidance. Dynavect's contribution targets feed-forward latent-space modulator for StyleGAN. We nonetheless included two diffusion editors(SD-Img2Img which was published in 2023 & InstructPix2Pix) to bracket pixel-space performance. We will be happy to add a small DiffusionCLIP comparison on our public cohort in the revision. It will not affect our core claim about context aware latent deltas + identity projection.
>
> 2. Reduction of entanglement quantitatively
> We will include a targeted measurement: per-image delta(attribute) from a frozen classifier (e.g., gender for "blonde", age for "age+") and report mean delta(CI) across seeds. Our pilot analyses indicate reduced drift vs. StyleCLIP-G when identity orthogonalization is enabled.

---

### Decision · Action_Editor_DBML · 2025-11-15

**Recommendation:** Reject

**Additional Comments:**

The submission presents a promising hybrid framework with practical insights into balancing edits and preserving identity in GAN-based editing, supported by user studies and ablations. However, it falls short of TMLR's standards for rigor and transparency due to inconsistent quantitative evidence, outdated baselines, vague reproducibility details, and presentation ambiguities (e.g., unclear motivation for hybrid design, confusing notation in Section 3.3). A major revision addressing these could elevate it to acceptance.

To strengthen the submission for resubmission:

Enhance Evidence and Comparisons (Critical): Add baselines against recent methods (e.g., DiffusionCLIP, InstructPix2Pix, StyleGAN3 mappers) and expand to non-facial domains (e.g., StyleGAN-Ada for cars) plus real images via inversion (e.g., PTI/e4e). Conduct rigorous metric analysis (e.g., Spearman correlations with user preferences) to substantiate claims on evaluation shortcomings. Quantify biases/entanglement (e.g., via classifiers for gender/ethnicity shifts) in failure cases (Figs. 9-11).

Improve Reproducibility and Details (Critical): Specify full training setup (dataset size, steps beyond 2000, exact losses/hyperparameters). Evaluate the Adaptive Loss Balancer (ALB) in results to justify its inclusion. Ablate key components like α and additional protected attributes (e.g., age/gender combos). Clarify optimization process in Section 3.5 (e.g., define "ideal" latent, address high-variance loss curves in Fig. 6).

Refine Presentation and Motivation (Critical): Strengthen Section 3.3 to clearly link orthogonal weight properties to entanglement reduction, with unambiguous notation (e.g., resolve ∆w inconsistencies) and a pipeline diagram. Articulate the hybrid motivation: e.g., global directions enable strong edits but entangle attributes; DCM personalizes corrections to mitigate this. Restructure Fig. 2 for clarity. Fix typos (e.g., future-dated refs like Guo et al. 2025).

**Audience:**

Yes

**Audience Explanation:**

TMLR's audience, spanning machine learning, generative models, and vision-language tasks, would find value in the hybrid editing approach (combining global directions with a context-aware modulator), the optimization-distillation technique for approximating per-image edits, and discussions of evaluation pitfalls in semantic editing (e.g., metric biases toward minimal changes). These elements could appeal to researchers in controllable generation, latent space manipulation, and knowledge distillation, particularly for legacy GAN applications like StyleGAN in resource-constrained settings. The user-study insights on perceptual preferences and critiques of attribute entanglement align with ongoing debates in ethical/fair editing. However, interest may be tempered by the reliance on dated models (StyleGAN2/CLIP ViT-B/32 from 2020-2021), as the field has pivoted toward diffusion models, and the narrow scope (faces only) limits broader applicability.

**Claims And Evidence:**

No

**Claims Explanation:**

The submission's core claims—regarding DynaVect's improved balance of edit strength and identity preservation, reduced attribute entanglement via orthogonal projection, and the efficiency of optimization-distillation training—are only partially supported. While qualitative results and user preference studies provide some visual and perceptual backing, quantitative evidence is inconsistent and unconvincing: DynaVect underperforms baselines like StyleCLIP-O on key metrics such as LPIPS, L2, and identity preservation in several cases, with modest ΔCLIP gains that do not robustly demonstrate the claimed trade-off. Claims about metric limitations (e.g., LPIPS favoring under-editing) remain anecdotal without rigorous analysis, such as correlation studies between metrics and user votes. The orthogonalization component shows ablation benefits but lacks deeper explanation of its mechanism or failure modes. Training details are vague (e.g., unspecified dataset size for distillation, only 2000 steps noted), hindering reproducibility, and the Adaptive Loss Balancer (ALB) is described but unused in results, leaving its purported benefits unsubstantiated. Comparisons are limited to outdated baselines (StyleCLIP, 2021), with no evaluation against recent diffusion-based (e.g., DiffusionCLIP, InstructPix2Pix) or advanced GAN methods, weakening novelty and generalizability claims. Experiments are confined to synthetic FFHQ faces with a narrow prompt set, omitting non-facial domains, real-image inversion, or complex prompts.

**Resubmission Of Major Revision:**

The authors may consider submitting a major revision at a later time.